# Phase-Selective Synthesis of Mo–Ta–C Ternary Nanosheets by Precisely Tailoring Mo/Ta Atom Ratio on Liquid Copper

**DOI:** 10.3390/nano12091446

**Published:** 2022-04-24

**Authors:** Rong Tu, Hang Yang, Chitengfei Zhang, Baowen Li, Qingfang Xu, Qizhong Li, Meijun Yang, Song Zhang

**Affiliations:** 1State Key Laboratory of Advanced Technology for Materials Synthesis and Processing, Wuhan University of Technology, Wuhan 430070, China; turong@whut.edu.cn (R.T.); deepyangsir@gmail.com (H.Y.); bwli@whut.edu.cn (B.L.); xuqingfanghed@gmail.com (Q.X.); qizhongli@whut.edu.cn (Q.L.); liyangmeijun@163.com (M.Y.); kobe@whut.edu.cn (S.Z.); 2Chaozhou Branch of Chemistry and Chemical Engineering Guangdong Laboratory, Chaozhou 521000, China; 3Wuhan University of Technology Advanced Engineering Technology Research Institute of Zhongshan City, Zhongshan 528400, China

**Keywords:** phase-selective synthesis, Mo–Ta–C ternary nanosheets, chemical vapor deposition, Mo/Ta ratio

## Abstract

Phase-selective synthesis is an effective way to expand the ultra-thin transition metal carbide family and tune its properties. Herein, a chemical vapor deposition route with specially designed substrate (Ta wire–Cu foil–Mo foil) is carried out to synthesize Mo–Ta–C ternary nanosheets with tunable phase structure. The Ta atoms diffuse on the surface of liquid copper and Mo atoms diffuse through the liquid copper to the surface, which react with the carbon atoms decomposed from the methane and form the Mo–Ta–C ternary nanosheets. By precisely tailoring the Mo/Ta ratio and growth temperature, ultrathin layered orthorhombic (Mo_2/3_Ta_1/3_)_2_C nanosheets and non-layered cubic (Mo_0.13_Ta_0.87_) C nanosheets with thickness of 21 and 4 nm are selectively synthesized. The approach could pave the way for the formation of multi-component carbide nanosheets with controllable phases.

## 1. Introduction

2D transition metal carbides (TMCs) have drawn great attention all over the world in the past few years due to the diverse physical and chemical properties, including superconductivity [1], valley polarization [2], and high catalytic activity [3,4]. Various binary TMC nanosheets (V_2_C [5], Ti_2_C [6], Mo_2_C [7,8] Nb_4_C_3_ [9], Zr_3_C_2_ [10], etc.) synthesized by liquid exfoliation or chemical vapor deposition (CVD) have been applied in catalysis, solar cells, and microelectronics, indicating that the synthesis and property research of binary TMCs tends to be mature.

In order to tune the properties of 2D TMCs further, defects, dangling bonds, and alloy elements are introduced (Appendix A) [11,12,13]. However, the introduction of defects and dangling bonds could tune the performance slightly and weaken the intrinsic performance of binary carbides significantly [14,15]. While the introduction of alloy elements leads to improving the performance fundamentally [16,17]. For example, the microhardness of the ternary carbides (Ti_0.4_Hf_0.6_)C is almost twice that of HfC [16]. Mo_2_TiC_2_ and Mo_2_Ti_2_C_3_ reveal substantially higher electrochemical behavior compared to Mo_2_C [17]. The synthesis of binary TMC nanosheets by CVD methods has made some progress in the past few years. Zhang et al. [18] directly obtained the self-aligned vanadium carbide with face-centered cubic structure on a unique stacked substrate by a CVD method. Xu et al. [1] achieved the layered hexagonal α-Mo_2_C and non-layered face-centered cubic WC on liquid copper by atmospheric pressure CVD, indicating the possibility of phase-selective synthesis for TMCs. However, limited by the synthesis strategies, it remains difficult to synthesize the ternary 2D TMC nanosheets by CVD method so far.

Herein, a sandwich-like substrate consisting of a Ta wire, Cu foil, and Mo foil is proposed to synthesize ternary 2D Mo–Ta–C (MTC) nanosheets. The Ta atoms diffuse on the liquid copper from the Ta wire and Mo atoms diffuse through the liquid copper from the bottom Mo foil. Followed by the introduction of CH_4_, C, Ta, and Mo atoms are bonded together and form the 2D MTC nanosheets on liquid copper. Ultrathin layered orthorhombic (Mo_2/3_Ta_1/3_)_2_C nanosheets and non-layered cubic (Mo_0.13_Ta_0.87_)C nanosheets were selectively synthesized into triangular and polygonal geometries by adjusting the molar ratio of Mo/Ta.

## 2. Materials and Methods

### 2.1. Synthesis of Single-Crystalline (Mo_2/3_Ta_1/3_)_2_C and (Mo_0.13_Ta_0.87_)C

The self-made chemical vapor deposition system is utilized to synthesize carbides. A molybdenum foil (20 × 20 mm, Innochem, Beijing, China 99.99%) 50 μm in thickness, Tungsten foil (20 × 20 mm, Innochem, Beijing, China 99.99%) 50 μm in thickness, a copper foil (20 × 20 mm, Innochem, Beijing, China 99.99%) 50 μm in thickness, and a tantalum wire (20 mm, Innochem, Beijing, China 99.99%) 0.5 mm in diameter are cleaned with acetone (Innochem, Beijing, China 95%), ethanol (Innochem, Beijing, China 95%), and deionized water. The Mo foil, Cu foil, Ta wire, and W foil are stacked from the top in the order of (a) Mo wire–Cu foil–Ta foil; (b) Mo wire and Ta wire–Cu foil–W foil; (c) Ta wire–Cu foil–Mo foil. After that, the furnace (Tianjin Zhonghuan Lab Furnace Co., Ltd., Tianjin, China) is heated to 1090–1170 °C within 90 min under the flow of Ar (300 sccm, 99.999%) and H_2_ (300 sccm, 99.999%) and held there for 30 to 60 min. Finally, with introducing the methane flow (0.1–2 sccm, 99.999%), graphene, TaC, and single crystalline (Mo_2/3_Ta_1/3_)_2_C and (Mo_0.13_Ta_0.87_)C phases were obtained on the liquid copper substrates under the different Mo/Ta ratios.

### 2.2. Transfer Process

A thin layer of poly (methyl methacrylate) (PMMA, weight-averaged molecular mass M_W_ = 600,000, 6 wt% in anisole) was first spin-coated on samples at 3000 rpm for 1 min and cured at 120 °C for 10 min. The PMMA-coated samples were then immersed in a 0.2 M (NH_4_)_2_SO_4_ solution for 24 h to etch the Cu substrate. Finally, the PMMA-coated ultrathin MTC crystals were stamped onto target substrates, such as SiO_2_/Si and TEM grids, and warm acetone (55 °C) was utilized to dissolve the PMMA layer and obtain clean ultrathin MTC crystals.

### 2.3. Characterization

Optical microscopy (OM, MX6RT, Sunny Optical Technology Co., Yuyao, China) is carried out in a reflection mode. The Bruker Multimode 8 DI (Burker, Billerica, MA, USA) in contact mode is performed for atomic force microscopy (AFM) imaging. Horiba Lab Ram HR (Horiba, Kyoto, Japan) 800 eV with 532 nm laser is performed for Raman analysis and imaging. A JEM-2100 F (JEOL, Akishima, Japan) at 200 kV is performed for high-resolution transmission electron microscopy (HRTEM) and selected-area electron diffraction (SAED).

## 3. Results

The phase diagrams of Cu-Mo and Cu-Ta are shown in Appendix A, indicating that the Cu cannot form intermetallic compounds with Ta and Mo [19,20]. Thus, the pure Ta and Mo wire, foil, or block could diffuse on liquid copper surface without damaging the atomic-level flat surface. When the temperature rises up to 1090 °C, Ta and Mo atoms would diffuse on or though the liquid copper and bond with carbon atoms decomposed from CH_4_. Several different stacking orders have been conducted: (a) Mo wire–Cu foil–Ta foil (Figure 1a); (b) Mo wire and Ta wire–Cu foil–W foil (Figure 1d); (c) Ta wire–Cu foil–Mo foil (Figure 1g).

For the order (a), EDS measurements indicate (Appendix A) no Mo and Ta was identified in the detection limit, and only graphene was obtained on the liquid copper (Figure 1b,c). For the order (b), W foil acts as support for liquid copper and triangular tantalum carbide nanosheets were achieved (Figure 1e,f), indicating that Ta could effectively diffuse on the surface of liquid copper and Mo was on the opposite. Based on the above results, the order (c) was carried out. Polygonal (Figure 1h) and triangular (Figure 1i) nanosheets were grown separately under the diffusion time of 30 and 60 min. The schematic diagram of the growth process is shown in Appendix A.

## 4. Discussion

In order to further identify the crystal structure and chemical composition of the two kinds of nanosheets, HRTEM, SAED, and EDS (Figure 2) are performed. Figure 2b–d are the EDS mapping of the crystal in Figure 2a, indicating that the crystal is composed of Ta, Mo, and C. The SAED pattern (Figure 2e) collected from Figure 2a is referred to the simple orthorhombic form (JCPDS no. 31-0871) with a space group of Pbcn(60), suggesting the chemical formula is (Ta_x_Mo_1−x_)_2_C. The atomic resolution image (Figure 2f) is observed under the [100] zone axis. The two different planes with the lattice spacing of 0.263 and 0.258 nm is consistent with the (021) and (002) plane. In order to confirm the ratio of Mo to Ta precisely, EDS point scan is performed at the nine selected points marked in Figure 2a. The ratio of Mo to Ta collected from the nine points is approximately 2:1, demonstrating the chemical formula is (Ta_1/3_Mo_2/3_)_2_C. Figure 2i is the atomic model of a unit cell of (Ta_1/3_Mo_2/3_)_2_C with the lattice parameters of *a* = 0.4754 nm, *b* = 0.5241 nm, *c* = 0.6076 nm, *α* = 90°, *β* = 90°, *γ* = 90°.

Figure 3a shows a low-magnification high-angle annular dark-field (HAADF)-STEM image of a triangular nanosheet crystal. Mo, Ta, and C are uniformly distributed across the whole sheet (Figure 3b–d), indicating the uniform chemical composition. Figure 3e shows the SAED pattern along the [1-1-1] zone axis. The two different planes with the lattice spacing of 0.157 and 0.090 nm are consistent with the (220) and (422) plane. These lattice parameters are consistent with cubic TaC (JCPDS no. 35-0801). EDS point scan is performed at the six selected points marked in Figure 3a. The ratio of Mo to Ta collected from the six points is approximately 13:87, demonstrating the chemical formula is (Ta_0.13_Mo_0.87_)C. Figure 3i is the atomic model of a unit cell of (Ta_1/3_Mo_2/3_)_2_C with the lattice parameters of *a* = 0.4455 nm, *b* = 0.4455 nm, *c* = 0.4455 nm, *α* = 90°, *β* = 90°, *γ* = 90°.

As shown in Raman spectra (Figure 4a), the orthorhombic MTC phase has two prominent peaks at the peak positions of 143 and 649 cm^−1^. The cubic MTC phase has two characteristic peaks located at 130 and 632 cm^−1^ in Figure 4e, respectively. The peak position at 130 cm^−1^ is attributed to the Ta-C bond, and the 143 cm^−1^ peak is related to the typical Mo-C bond of B_3g_ Raman modes [21]. The peak near 632 cm^−1^ is a typical peak of TMCs, as seen at NbC [22], TaC [23], and Mo_2_C [21,24]. Meanwhile, Figure 4c,g depict the Raman mappings of the orthorhombic and cubic phases, demonstrating the uniformity of the two phases. The size of polygonal nanosheet is 84 μm (in Figure 1f) in length and 21 nm in thickness (Figure 4d). Another phase is typically triangular with a size of 15 μm and thickness of 4 nm (Figure 4h).

Figure 5 shows the effect of temperature and Mo/Ta atomic fraction on the growth of orthorhombic and cubic MTC. The Mo/Ta ratio is controlled by the diffusion time.

Both orthorhombic and cubic MTC nanosheets were synthesized at 1090–1170 °C. Orthorhombic MTC nanosheets were obtained at Mo/Ta atomic fraction of Ta from 10% to 40%. Cubic MTC nanosheets were obtained at Mo/Ta atomic fraction of Ta from 60% to 90%. Mo_2_C nanosheets were obtained (Appendix A) without the Ta wire. The corresponding optical images are shown in Appendix A. When the atomic percentage of Ta was low (10–20%), the orthorhombic MTC nanosheets tended to form larger single nanosheets with lower nucleation density (Appendix A). When the atomic percentage of Ta was high (80–90%), cubic MTC nanosheets tended to aggregate together (Appendix A).

## 5. Conclusions

A phase selective CVD method with the special stacked order of Ta wire–Cu foil–Mo foil was developed to control the synthesis of orthorhombic and cubic MTC nanosheets with high crystallinity on liquid copper. The maximum size of orthorhombic MTC nanosheets reached 84 μm grown at Mo/Ta ratio of 9:1. While the cubic MTC nanosheets had the largest size of 18 μm grown at Mo/Ta ratio of nearly 1:9, demonstrating that the growth of MTC is deeply influenced by the Mo/Ta molar ratio. This study paves a new way for the synthesis of multiphase 2D nanosheets with controllable phase.

## Figures and Tables

**Figure 1 nanomaterials-12-01446-f001:**
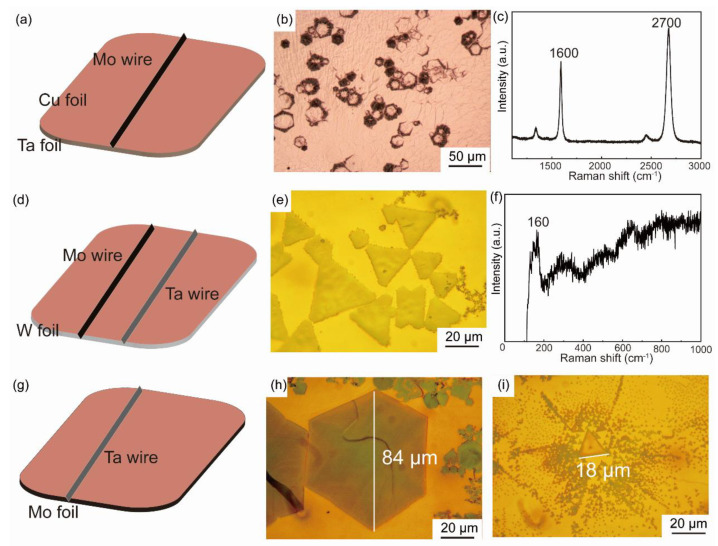
Several different stacking orders of transition metallic source: (**a**) Mo wire–Cu foil–Ta foil, (**b**) optical image and (**c**) Raman spectrum of graphene corresponding to (**a**). (**d**) Mo wire and Ta wire–Cu foil–W foil, (**e**) optical image and (**f**) Raman spectrum of TaC corresponding to (**d**). (**g**) Ta wire–Cu foil–Mo foil, (**h**) optical image of polygonal and (**i**) triangular nanosheet corresponding to (**g**).

**Figure 2 nanomaterials-12-01446-f002:**
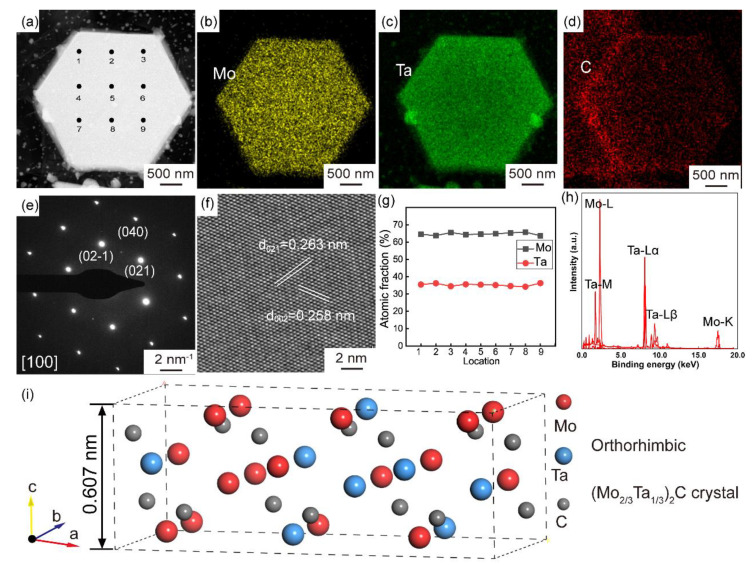
TEM images of polygonal MTC crystal (**a**) and corresponding EDS elemental mapping of Mo (**b**), Ta (**c**), and C (**d**). SAED pattern along [100] zone axis (**e**) and HRTEM image of a polygonal crystal (**f**). Atomic fraction of Mo and Ta in the polygonal crystal of (**a**). (**g**) Based on EDS spectrum (**h**). (**i**) Crystallographic structure of orthorhombic (Mo_2/3_Ta_1/3_)C.

**Figure 3 nanomaterials-12-01446-f003:**
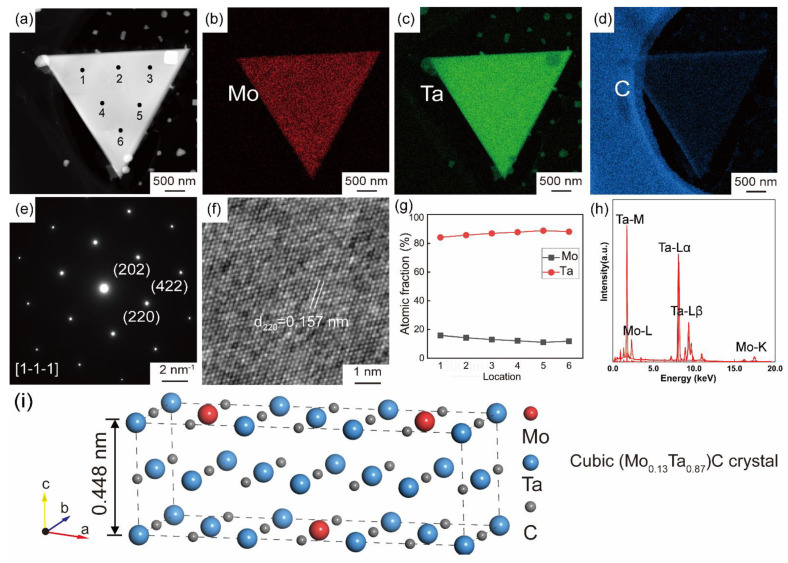
TEM images of triangular MTC crystal (**a**) and corresponding EDS elemental mapping of Mo (**b**), Ta (**c**), and C (**d**). SAED pattern along [1] zone axis (**e**) and HRTEM image of a triangular crystal (**f**). Atomic fraction of Mo and Ta in the triangular crystal of (**a**). (**g**) Based on EDS spectrum (**h**). (**i**) Crystallographic structure of cubic (Mo_0.13_Ta_0.87_)C.

**Figure 4 nanomaterials-12-01446-f004:**
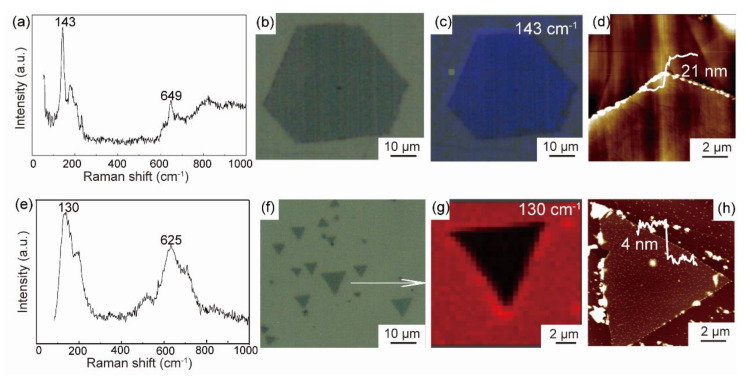
Raman spectra (**a**,**e**), optical images (**b**,**f**), Raman mapping (**c**,**g**) and AFM images (**d**,**h**) of orthorhombic (**a**–**d**) and cubic (**e**–**h**) MTC phases.

**Figure 5 nanomaterials-12-01446-f005:**
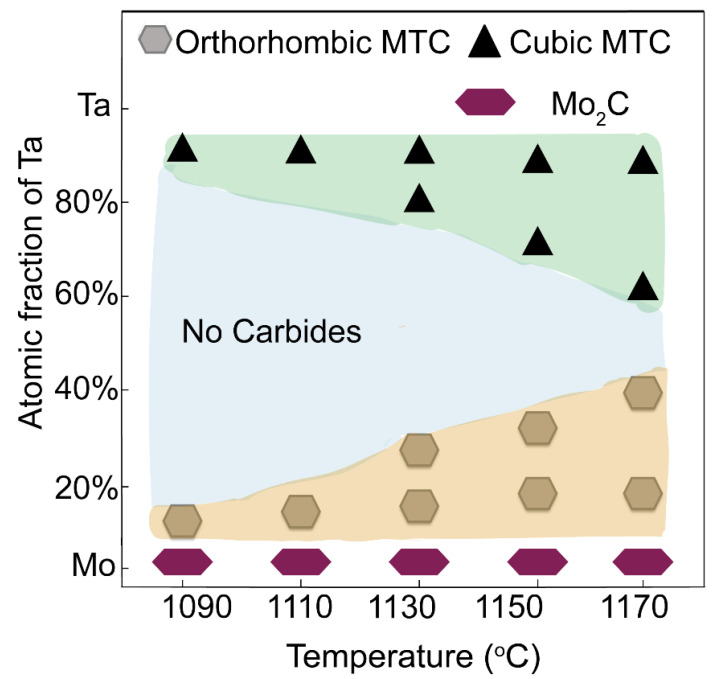
Effect of temperature and Mo/Ta atomic fraction on the growth of orthorhombic and cubic MTC.

## Data Availability

Data can be available upon request from the authors.

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
