# Peer review of "Phase-Selective Synthesis of Mo–Ta–C Ternary Nanosheets by Precisely Tailoring Mo/Ta Atom Ratio on Liquid Copper"

_nanomaterials, 2022, doi:10.3390/nano12091446_

Round 1
Reviewer 1 Report
The authors reported on the use of CVD method with designed substrate (Ta wire-Cu foil-Mo foil) to grow Mo-Ta-C ternary nanosheets with tunable phase structure. In general the paper is well-written and presents interesting results concern the synthesis of Mo-Ta-C ternary nanosheets. Some suggestions are addressed before paper acceptance:
(i) The legend of Figure 1 presents the figures out of order, please put the title of the images in order (a), (b), (c) ....
(ii) Concern the phrase: "... is heated to 1090ºC within 90 min under the flow of Ar..." , the correct is "1090 to 1170ºC"?
Reviewer 2 Report
In the manuscript titled "Phase-Selective Synthesis of Mo-Ta-C Ternary Nanosheets by Precisely Tailoring Mo/Ta Atom Ratio on Liquid Copper," Tu et al used CVD method to grow binary metal carbide sheets. However, the work is not presented properly and the synthesis procedure is hard to understand. The language needs extensive improvement. Here are only a few of the specific comments.
- A schematic of the synthesis procedure should be properly presented to understand the mechanism involved.
- What is the thermodynamics behind the synthesis procedure? At the working temperature 1090 degrees C (40 degrees larger than the copper melting point) the copper will be in the liquid phase, but Ta (MP 3020) and Mo (MP 2623) remain in the solid phase. How do the atoms from the solid Mo and Ta leave their corresponding surfaces and go to the liquid copper? Now if copper is completely going to liquid phase and Mo and Ta diffuses into liquid copper from two opposite sides, then which copper substrate needs to etch out.
- the left column in Fig. 1 shows different arrangements of metal wires and foils to synthesize different morphologies of TMCs, but these are not discussed in the experimental section.
- Several statements are made without proper citations. For example : "However, the introduction of defects and dangling bonds could tune the performance slightly and weaken the intrinsic performance of binary carbides significantly." needs citations. Similarly, the sentence, "While alloy elements are introduced into the transition metal site of 2D TMCs, leading to improving the performance fundamentally." needs citations. This problem is throughout the manuscript.
- To present the motivation of the work, authors wrote " However, limited by the synthesis strategies, the ternary 2D TMC nanosheets
have not been obtained by CVD method so far." But what about TMCs by other methods. Why CVD is so important? Is CVD is the motivation of the work or TMC?
